# Aliphatic Quaternary Ammonium Functionalized Nanogels for Gene Delivery

**DOI:** 10.3390/pharmaceutics13111964

**Published:** 2021-11-19

**Authors:** Huaiying Zhang, Damla Keskin, Willy H. de Haan-Visser, Guangyue Zu, Patrick van Rijn, Inge S. Zuhorn

**Affiliations:** Department of Biomedical Engineering, University Medical Center Groningen, University of Groningen, 9713 AV Groningen, The Netherlands; h.zhang@umcg.nl (H.Z.); d.keskin@umcg.nl (D.K.); h.w.de.haan-visser@umcg.nl (W.H.d.H.-V.); g.zu@umcg.nl (G.Z.)

**Keywords:** nanogels, gene delivery, endosomal escape, quaternization, aliphatic chains

## Abstract

Gene therapy is a promising treatment for hereditary diseases, as well as acquired genetic diseases, including cancer. Facing the complicated physiological and pathological environment in vivo, developing efficient non-viral gene vectors is needed for their clinical application. Here, poly(*N*-isopropylacrylamide) (p(NIPAM)) nanogels are presented with either protonatable tertiary amine groups or permanently charged quaternized ammonium groups to achieve DNA complexation ability. In addition, a quaternary ammonium-functionalized nanogel was further provided with an aliphatic moiety using 1-bromododecane to add a membrane-interacting structure to ultimately facilitate intracellular release of the genetic material. The ability of the tertiary amine-, quaternized ammonium-, and aliphatic quaternized ammonium-functionalized p(NIPAM) nanogels (i.e., NGs, NGs-MI, and NGs-BDD, respectively) to mediate gene transfection was evaluated by fluorescence microscopy and flow cytometry. It is observed that NGs-BDD/pDNA complexes exhibit efficient gene loading, gene protection ability, and intracellular uptake similar to that of NGs-MI/pDNA complexes. However, only the NGs-BDD/pDNA complexes show a notable gene transfer efficiency, which can be ascribed to their ability to mediate DNA escape from endosomes. We conclude that NGs-BDD displays a cationic lipid-like behavior that facilitates endosomal escape by perturbing the endosomal/lysosomal membrane. These findings demonstrate that the presence of aliphatic chains within the nanogel is instrumental in accomplishing gene delivery, which provides a rationale for the further development of nanogel-based gene delivery systems.

## 1. Introduction

Gene therapy represents a promising approach for the treatment of a wide variety of human diseases, such as Duchenne muscular dystrophy, Parkinson’s disease, Huntington’s disease, and cancers [1,2,3,4]. Despite significant progress, a lack of efficient and safe gene delivery vectors is still the primary obstacle in gene therapy. The most extensively used delivery vectors are viral vectors, which enable the efficient delivery of genes. However, concerns with the use of viruses, including the high production cost, and risk of immunogenicity and induced oncogenic transformations have led researchers to search for non-viral gene vectors [5,6]. Until now, cationic polymers (e.g., poly(ethyleneimine), polylysine, and cationic polysaccharides) [7,8,9] and cationic lipids (e.g., Lipofectin^TM^ and LipofectAMINE^TM^) [10,11,12,13] were the most investigated classes of synthetic materials for gene delivery. Cationic polymers usually contain primary, secondary, and/or tertiary amino groups and provide a positive charge through the presence of protonated amines, which mediates complex formation with anionic gene molecules through electrostatic interactions [14]. While cationic lipids bearing a quaternary ammonium group are successfully used for in vitro gene delivery, quaternary ammonium cationic polymers generally show reduced transfection efficiency and increased cytotoxicity [15,16]. This can be explained by the different mechanisms by which cationic polymers and cationic lipids induce the endosomal escape of genetic cargo. Specifically, additional protonation upon a drop in pH, as occurs during endosomal maturation, has been held responsible for the ability of cationic polymers that contain protonatable groups to induce the endosomal escape of genetic cargo and resulting transfection of cells via the so-called proton sponge effect, i.e., osmotic rupture of endosomes [17,18,19]. In sharp contrast, cationic lipids induce transient pores in the endosomal membrane, which allows for the release of genes into the cell cytosol. This pore formation is dependent on the formation of non-lamellar lipid phases and is pH-independent [19,20,21]. The recently developed protonatable/ionizable cationic lipids in lipid nanoparticle (LNP) formulations display a similar mechanism of endosomal escape as non-ionizable cationic lipids. Their main advantage is their charge neutrality during circulation in the blood, promoting long circulation half-life after IV administration [22].

Overall, it is difficult to flexibly adjust and control the properties of the complexes formed by traditional cationic lipids or polymers and genes, such as the particle size, surface charge, and gene release from the complex, to meet the requirements for successful in vivo gene transfection.

Among the variety of non-viral vectors developed for gene delivery, nanogels have aroused great interest because of their excellent payload capacity, biocompatibility, and ease of functionalization [23,24,25,26,27,28]. Nanogels are nanoparticles that consist of a cross-linked and porous hydrophilic polymer network, which can be designed to respond to various stimuli (e.g., temperature, pH, and light) and can be functionalized to introduce various functional groups [29,30,31,32]. So far, a variety of nanogels has been used for catalysis [33,34], selective diagnostics and delivery [35,36,37], and anti-bacterial and antifouling coatings [38,39]. Among them, the thermoresponsive poly(*N*-isopropylacrylamide) p(NIPAM) nanogel undergoes a swollen-to-collapsed transformation at the volume phase transition temperature (VPTT, ~32 °C) [32]. Even though p(NIPAM) nanogels have been widely exploited for biomedical applications in the form of dispersed particles, 2D films, and 3D aggregates [40], including siRNA delivery carriers [41,42,43], few DNA delivery vectors have been developed that rely solely on hydrogel structures.

Recently, our group reported that a p(NIPMAM) nanogel with low stiffness showed efficient transcytosis across an in vitro blood-brain barrier [44]. In addition, introducing quaternary amine compounds (QACs) with long-chain alkyl groups (12 carbons) into the pNIPAM nanogels enabled efficient delivery of a hydrophobic antimicrobial by direct interaction with the bacterial membrane through intercalation [45,46]. We hypothesized that a similar process may occur within endosomal membranes, which may provide the nanogel with the property to trigger endosome permeabilization and mediate gene transfection. Therefore, here, our aim was to investigate the gene delivery potential of QAC-pNIPAM nanogels bearing aliphatic membrane interacting groups, as compared to tertiary amino groups and QACs without the aliphatic character. For this purpose, a tertiary amine-functionalized poly(*N*-isopropylacrylamide*-co-N*-[3-(dimethylamino)-propyl] methacrylamide) (p(NIPAM*-co-*DMAPMA) nanogel (NGs) was synthesized. Subsequently, the NGs were quaternized by *N*-alkylation with methyl iodide (NGs-MI) or 1-bromododecane (NGs-BDD). Our data show that NGs, NGs-MI, and NGs-BDD nanogel efficiently bound pDNA at a 10:1 weight ratio and protected the DNA from degradation by serum nucleases. All three nanogel/pDNA complexes were efficiently internalized by HEK-293T cells, while only NGs-BDD/pDNA complexes showed the ability to induce endosomal escape of the DNA and subsequent cell transfection. The endosomal escape and cell transfection induced by NGs-BDD was pH-independent and relied on the membrane-disruptive action of the BDD chains.

## 2. Materials and Methods

### 2.1. Materials

2,2′-Azobis(2-methylpropioamidine)dihydrochloride (AMPA, V50, 97%), *N*-[3-(dimethylamino)propyl]methacrylamide (DMAPMA, 99%), N,N′ methylene-bis(acrylamide) (BIS, 99%), potassium carbonate (K_2_CO_3_), hexadecyltrimethylammonium bromide (CTAB, 99%), *N*,*N*-dimethylformamide (DMF, anhydrous), methyl iodide, polyethylenimine (PEI, branched, Mw 25,000 g/mol), 1-bromo-dodecane (97%), and deuterium oxide (D_2_O) were obtained from Sigma-Aldrich (Darmstadt, Germany). Lipofectamine 2000 was obtained from ThermoFisher Scientific, Invitrogen (Carlsbad, CA, USA). *N*-isopropylacrylamide (NIPAM, 98%) was obtained from TCI (Zwijndrecht, Belgium). Potassium chloride (KCl), ethanol, methanol (anhydrous), and tetrahydrofuran (THF, anhydrous) were obtained from Merck, (Darmstadt, Germany). All chemicals were used as received without any further purification. Opti-MEM, Dulbecco’s modified eagle’s medium (DMEM, high glucose), trypsin, fetal bovine serum (FBS), 3-(4,5-dimethylthiazol-2-yl)-2,5-diphenyltetra-zolium bromide (MTT), and Hoechst33324 were obtained from Thermo FisherScientific (Waltham, MA, USA). Lysotracker green DND-26 was from ThermoFisher Scientific Invitrogen (Carlsbad, CA, USA). The reporter plasmid pEGFP-N1 was amplified in *Escherichia coli* DH5-a and then purified using a QIAGEN Plasmid Maxi kit (QIAGEN GmbH, Hilden, Germany). The plasmid was labeled with the fluorophore Cy5 using the Mirus Label IT kit (Mirus Bio™, Madison, WI, USA). TAMRA-labeled oligodeoxynucleotides (ODNs) (TAMRA-5′-AC- TACTACACTAGACTAC-3′) were from Merck (Darmstadt, Germany).

### 2.2. Instruments and Methods

Proton nuclear magnetic resonance (NMR) spectra were recorded on a Varian Mercury-400 NMR spectrometer (400 MHz) in D_2_O using tetramethylsilane (TMS) as a reference. The particle sizes and zeta potentials of complexes at weight ratio 10 were determined using a commercial laser light scattering instrument (Malvern ZEN3690, Malvern Panalytical Ltd., Cambridge, UK) at room temperature. The hydrodynamic diameter of nanogels was measured from 25–80 °C with 2 °C intervals [45]. The morphologies of nanogels were characterized through atomic force microscopy (AFM) imaging (Dimension 3100 Nanoscope V, Veeco, Plainview, NY, USA). DNP cantilevers (k = 0.24 N/m) were operated in the air using contact mode.

### 2.3. Synthesis of the Nanogels

#### 2.3.1. Synthesis of the Non-Quaternized Nanogel (NGs)

The P(NIPAM*-co-*DMAPMA) was synthesized through precipitation polymerization, as previously reported [47]. In total, 3.37 g of NIPAM (29.8 mmol, 85 mol %), 0.27 g of BIS (1.7 mmol, 5 mol %), and 0.0106 g of CTAB (0.03 mmol) were dissolved in ultrapure water (236 mL) in a three-necked flask, together with a reflux condenser and a magnetic stirrer. After degassing by bubbling N_2_ through the solution for 1 h, the mixture was heated to 85 °C. Subsequently, 10 mL of degassed DMAPMA solution (0.6 g, 3.4 mmol, 10 mol %) was added via a syringe. The pH was adjusted to 8–9 with 0.1 M degassed NaOH and HCl solutions. Then, 14 mL of degassed AMPA V50 solution was injected (0.135 g, 0.5 mmol) for initiation of the reaction and stirred at 300 rpm for 6 h at 85 °C under an N_2_ atmosphere. After 6 h of reaction, the solution was cooled to room temperature, and stirring was continued overnight. Purification of the nanogel dispersion was done by dialysis against water for 3 days (MWCO 3500 Da, Spectrum Laboratories Inc., Rancho Dominguez, CA, USA). The purified NGs were freeze-dried.

#### 2.3.2. Quaternization of Nanogel with Methyl Iodide (NGs-MI)

The previously synthesized NGs were quaternized with methyl iodide (MI), as previously reported [47]. Briefly, 0.4 g of nanogel and 0.058 g of K_2_CO_3_ were dispersed in 30.3 mL of methanol. The reaction was started after adding 0.17 mL (2.73 mmol) of MI and was kept stirring for 4 days at room temperature. Subsequently, water was added to the reaction mixture, and the methanol was removed under reduced pressure. After purification via dialysis against water for 3 days (MWCO 3500 Da), the product NGs-MI was freeze-dried for further use.

#### 2.3.3. Quaternization of Nanogel with 1-Bromododecane (NGs-BDD)

NGs was quaternized with 1-bromododecane, according to a previous report, with a minor change [48]. NGs (500 mg) and 82 mg of NaOH were dissolved in 100 mL of DMF in a round-bottom flask with a stirrer. In total, 1.027 g of 1-bromo-dodecane (BDD) was injected into the flask, and the reaction was started. The reaction was continued with stirring for 4 days at 80 °C. Impurities were removed via dialysis (MWCO 3500 kDa) against 96% ethanol for 3 days and afterward against water, also for 3 days. The pure NGs-BDD was isolated as a powder by freeze-drying.

### 2.4. Buffering Capacities

The proton buffering capacities of NGs, NGs-MI, NGs-BDD, and PEI-25kDa were measured by acid-base titration. In total, 10 mg of nanogel was dissolved in 30 mL of ultrapure water, and the initial pH was adjusted to 3.0 with 0.1M HCl. The solution was subsequently titrated to a pH of 11.0 with 0.1 M sodium hydroxide (NaOH), added in portions of 6 µL. The pH was monitored by a Metrohm combined glass electrode (SM Titrino702, Metrohm Netherlands, Barendrecht, The Netherlands) after the addition of each aliquot. Ultrapure water served as the control.

### 2.5. Gel Retardation Electrophoresis

Nanogel powders were dissolved in ultrapure H_2_O at a concentration of 1 mg/mL. Nanogel/pDNA complexes at weight ratios from 0–15 were freshly prepared by adding the right amount of nanogel in 10 µL ultrapure water to 0.3 µg of pDNA in 10 µL ultrapure water, followed by incubation at room temperature for 30 min. In total, 10 µL complex solutions were mixed with 2 µL of glycerol (30%) and loaded onto an agarose gel containing Midori Green Advance DNA stain (Nippon Genetics, Düren, Germany). For serum stability investigation, 10 µL complexes (prepared at weight ratio of 10) were incubated in 0, 10, or 50% FBS by adding 10 µL dd H_2_O, 20% FBS and 100% FBS, respectively. After 10 h, 10 µL complex solutions were mixed with 2 μL of glycerol and loaded onto the agarose gel. Gel electrophoresis was carried out in 1 × TAE buffer at 80 V for 30 min. The gel was imaged with a UVP GelDoc IT2 Imager (Analytik Jena GmbH, Upland, CA, USA).

### 2.6. Cytotoxicity Assay

HEK293T cells were cultured in DMEM-HG, containing 10% FBS and 1% penicillin–streptomycin at 37 °C/5% CO_2_. Cell viability was measured using the MTT colorimetric assay. Briefly, HEK293T cells were seeded at a density of 5 × 10^3^ cells/well in a 96-well plate and incubated for 24 h. Subsequently, the medium was replaced with fresh medium containing nanogels at different concentrations with further incubation for 48 h. In total, 50 μL of a 0.5 mg/mL solution of MTT reagent dissolved in PBS was added to each well. After 2 h incubation at 37 °C, the medium was replaced with 200 μL of dimethyl sulfoxide (DMSO) to dissolve the MTT formazan crystals. The Absorbance at 570 and 630 nm was measured in a microplate reader (SpectraMax M3, Molecular Devices LLC, San Jose, CA, USA) after shaking for 15 min. Untreated cells were used as control. Cell viability was calculated as described in the following equation: Cell viability (%) = (A_570Sample_ − A_630Sample_)/(A_570control_ − A_630control_) × 100, where A_570_ is the absorbance at 570 nm and A_630_ is the absorbance at 630 nm. For assessing the cytotoxicity of the complexes, HEK293T cells were treated with 0.5 mL opti-MEM containing nanogel complexes for 8 h, followed by incubation in fresh complete cell culture medium for another 40 h. Cell viability was measured using the same protocol as described above.

### 2.7. In Vitro Transfection

HEK293T cells at 10^5^ cells per well were seeded in 12-well plates in 1 mL cell culture medium and incubated for 24 h. Nanogel/pDNA complexes at weight ratio 10 were prepared by adding 10 µg nanogel in 50 µL ultrapure water to 1 µg of pEGFP-N1 in 50 µL ultrapure water, followed by gentle mixing and incubation at room temperature for 30 min. PEI/pDNA complexes at N/P ratio 10 were prepared by adding 1.3 µg PEI in 50 µL ultrapure water to 1 µg of pEGFP-N1 in 50 µL ultrapure water, followed by gentle mixing and incubation at room temperature for 30 min. Lipofectamine/pDNA complexes were prepared by adding 1 µg of pEGFP-N1 in 50 µL ultrapure water to 2 µL Lipofectamine in 50 µL ultrapure water, followed by gentle mixing and incubation at room temperature for 30 min. The medium of the HEK293T cells was aspirated and the cells were rinsed with PBS before exposure to 0.5 mL fresh opti-MEM containing complexes. After incubation for 8 h at 37 °C, the transfection medium was exchanged for 1 mL of cell culture medium and cells were incubated for another 40h. Transfected cells were visualized and photographed by fluorescence microscopy (Leica DMI6000B, Leica Microsystems Wetzlar GmbH, Wetzlar, Germany) using appropriate filters (Ex/Em: 485/520 nm) to qualitatively assess the green fluorescent protein (EGFP) expression. Transfected cells were then washed once with PBS and collected by trypsinization and centrifugation at 1000 rpm for 5 min. The cells were resuspended in 0.3 mL PBS. The transfection efficiency was quantitatively measured by a BD-LSR-II using a 488 nm laser, Blue-B detector (GFP: 530/30 filter) (Becton–Dickinson, San Jose, CA, USA). Cells pretreated with Bafilomycin A1 (300 nM) (MedChemExpress LLC, Princeton, NJ, USA) for 30 min, and subsequently exposed to complexes as described above, were assessed for transfection efficiency using a NovoCyte Quanteon Flow Cytometer using a 488 nm Laser, B530/30 filter (ACEA Biosciences, San Diego, CA, USA). Data analysis was performed using the NovoExpress V1.5 Software (Agilent Technologies, Santa Clara, CA, USA).

### 2.8. Cellular Uptake and Colocalization with Lysotracker

Circular glass coverslips (24 mm diameter) were sterilized by immersion in absolute ethanol overnight. They were transferred in 12-well plates and washed with PBS. HEK293T cells at 2 × 10^5^ cells per well were seeded in the plates and incubated overnight. In total, 1 µg of pDNA (containing 0.5 µg Cy5-labeled DNA) was complexed with nanogel at a weight ratio of 10. The cell medium was aspirated, and the cells were rinsed with PBS before exposure to 0.5 mL fresh opti-MEM containing complexes. After incubation for 1 h or 4 h at 37 °C, cells were rinsed three times with PBS and cell nuclei were stained with 2 µM Hoechst 33342 for 15 min at 37 °C. Fluorescence images were acquired with a confocal scanning laser microscope (CLSM, Leica TCS SP8, Leica Microsystems CMS GmbH, Mannheim, Germany). To determine colocalization of nanogel/pDNA complexes with lysotracker, after 4h incubation with complexes, cells were rinsed three times with PBS and incubated with 75 nM Lyso-Tracker green DND-26 for another 1 h, according to the manufacturer’s protocol. The nuclei were then stained with Hoechst 33342 (2 µM) for 15 min and samples were investigated with CLSM.

For quantitative evaluation of the cellular uptake of nanogel/pDNA complexes, cells were trypsinized after 1 h or 4 h incubation with complexes, centrifuged at 1000 rpm, and resuspended in PBS. The Cy5 fluorescence intensity in cells was measured using a 635 nm Laser and Red-C detector (Cy5: 660/20 filter) on the BD-LSR-II flow cytometer (Becton–Dickinson, San Jose, CA, USA). The mean fluorescence intensity (MFI) of cells was analyzed using FlowJo V10 software (FlowJo LLC, Ashland, OR, USA).

### 2.9. Endosomal Escape of ODNs Mediated by Nanogel/ODNs Complexes

HEK293T cells at 2 × 10^5^ cells per well were seeded in 12-well plates with circular glass coverslips and incubated overnight. In total, 0.1 nmol TAMRA-labeled ODNs were complexed with the same weight of nanogel as used to prepare complexes with 1 µg pDNA (see above). PEI-25kDa/ODNs complexes were prepared using 1.3 µg PEI-25kDa. The cell medium was aspirated, and the cells were rinsed with PBS before exposure to 0.5 mL fresh opti-MEM containing complexes. After incubation for 4 h at 37 °C, cells were rinsed three times with PBS and the nuclei were stained with Hoechst 33342 (2 µM) for 15 min. CLSM images were acquired. In total, 150–200 cells (9 images in total) per condition were counted to calculate the percentage of ODN-positive nuclei. Experiments were repeated three times. To investigate the influence of Bafilomycin A1 on the endosomal escape of ODNs, complexes were added to cells with or without Bafilomycin A1 pretreatment (30 min) and incubated for 4 h at 37 °C. After that, cells were rinsed three times with PBS, and cell nuclei were stained with 2 µM Hoechst 33342. Samples were investigated by CLSM.

### 2.10. Membrane Perturbing Activity of Nanogels and Nanogel/pDNA Complexes

The membrane perturbing activity of nanogels and nanogel/pDNA complexes was evaluated by using guinea pig erythrocytes (tebu-bio, Heerhugowaard, The Netherlands), according to the previous method [49]. In brief, 1 × 10^8^ guinea pig erythrocytes were added to 0.5 mL HBS containing nanogel (5 µg) or nanogel/pDNA complexes (0.5 µg pDNA) and incubated at 37 °C. After 60 min, the resulting mixtures were centrifuged at 2000× *g* at 4 °C for 5 min. The absorbance at 540 nm of the supernatant was measured in a microplate reader (SpectraMax M3, Molecular Devices LLC, San Jose, CA, USA) to quantify the extent of hemoglobin release. Erythrocytes incubated without nanogel were used as the negative control, and erythrocytes incubated with 1% (*v*/*v*) Triton X-100 were used as the positive control.

### 2.11. Statistics

The data were analyzed using GraphPad Prism 8 (GraphPad, La Jolla, CA, USA) and one-way ANOVA Tukey’s multiple comparison test, and Student’s t-test was used to determine statistical significance. All the data are representative of at least three independent experiments. *p* < 0.05 was considered to be statistically significant.

## 3. Results and Discussion

### 3.1. Preparation and Characterization of Nanogels

The nanogel (NGs) p(NIPAM*-co-*DMAPMA) was synthesized via precipitation polymerization with the thermoresponsive monomer NIPAM and comonomer DMAPMA, which contains a tertiary amine group. Afterward, the tertiary amines were alkylated by methyl iodide (MI) or 1-bromododecane (BDD), creating the quaternized nanogels NGs-MI and NGs-BDD with one and 12 carbon alkyl chains, respectively (Figure 1). Investigation by atomic force microscopy showed that all three nanogels, i.e., NGs, NGs-MI, and NGs-BDD, possessed a spherical shape and a homogeneous particle size (Figure 2A). The quaternization of the tertiary amine groups in the NGs-MI and NGs-BDD was firstly verified with ^1^H NMR spectroscopy. The 2.5 ppm peak in the NMR spectrum of the NGs depicts the methyl protons of DMAPMA. Disappearance of this signal in the spectra of the NGs-MI and NGs-BDD nanogels indicates their quaternized state (Appendix A). In addition, the formation of the quaternary ammonium compound in the nanogels was also confirmed by zeta potential measurement. The zeta potential of NGs showed a significant decrease from +28.4 to +12.5 mV, with increasing pH (from pH 4–11), whereas the zeta potential of NGs-MI and NGs-BDD was largely unaffected by changes in pH, indicating the permanent and pH-insensitive positive charges of quaternized NGs (Figure 2B).

As NIPAM is a thermosensitive polymer, with a volume-phase transition temperature (VPTT) of 32 °C, the hydrodynamic diameter of the NIPAM*-co-*DMAPMA nanogels at different temperatures was investigated by DLS (Figure 2C). Typically, at temperatures higher than the VPTT, the hydrodynamic diameter (d_h_) of pNIPAM nanogels decreases with increasing temperature [30]. The NGs showed a VPTT of ~38 °C, which was attributed to the incorporation of DMAPMA comonomer. The electrostatic repulsion of charged DMAPMA and increasing coordination of water molecules in the nanogel network altered the thermosensitivity, which led to the VPTT increase as compared to pure NIPAM [50,51]. After quaternization with MI and BDD, the thermoresponsiveness was less pronounced compared to that of the NGs, because of persistent electrostatic repulsion of quaternary amine moieties that hindered the hydrophobic collapse process.

Next, the proton buffering capacities of nanogels was evaluated by acid-base titration from pH 3.0–11.0 (Appendix A). The titration curve of NGs-MI and NGs-BDD is similar to that of H_2_O, indicating the absence of buffering capacity of both quaternized nanogels. The buffering capacity (β, β = dn(OH-)/dpH) of NGs with tertiary amines in the pH range between 4 and 7.4 was 1.76 × 10^−6^, about 181 times lower than that of branched PEI-25 kDa (0.32 × 10^−3^), indicating minimal proton buffering capacity of the NGs. This can be explained by the low number of amines in NGs due to a 11.4 mol% feed ratio of DMAPMA to NIPAM during NIPAM*-co-*DMAPMA synthesis.

### 3.2. Complexes of Nanogels and Plasmid DNA

To determine the DNA complexation capacity of nanogels, the retardation of DNA after incubation with nanogel was assessed by gel retardation assay. Complexes between nanogels and pDNA were prepared at different weight ratios (Figure 3A). Complete retardation of DNA with NGs and NGs-MI was observed at a weight ratio of 5 or higher, while NGs-BDD already completely retarded the migration of pDNA at a weight ratio of 2, demonstrating a higher DNA-binding capacity of NGs-BDD compared to NGs and NGs-MI. In addition, the average hydrodynamic diameter and zeta potential of the DNA-loaded nanogels (weight ratio of 10) was determined. As shown in Figure 3B, the size of NGs/pDNA, NGs-MI/pDNA, and NGs-BDD/pDNA complexes was 423 ± 14, 469 ± 18, and 476 ± 22 nm, respectively, which was comparable to that of the unloaded nanogels (cf. Figure 2C). DNA loading resulted in a decrease in the positive charge of nanogels, presenting complexes with a zeta potential around +6 to +10 mV (Figure 3C).

We further examined the stability of the nanogel/pDNA complexes in the presence of serum by means of gel electrophoresis. Nanogel/pDNA complexes were prepared at a weight ratio of 10, and DNA integrity was evaluated after incubation in dd H_2_O containing 10% or 50% FBS for 10 h. As shown in Appendix A, naked pDNA completely degraded when exposed to 10% serum. Unexpectedly, incubation of pDNA with 50% serum resulted in little degradation, presumably because of the binding of ‘protective’ serum components to the DNA, as can be appreciated from the impeded DNA migration into the agarose gel. Nanogel/pDNA complexes showed no signs of DNA degradation in the presence of 10 and 50% serum, as indicated by the absence of DNA signal loss in the sample slots, which shows that NGs/pDNA, NGs-MI/pDNA, and NGs-BDD/pDNA complexes protected the DNA from serum nuclease activity.

### 3.3. Cellular Uptake and Endosomal Escape of Nanogel/pDNA Complexes

For gene delivery applications, it is important that the gene vectors are non-toxic to cells. Therefore, the cytotoxicity of NGs, NGs-MI, NGs-BDD, and their complexes with pDNA (weight ratio of 10) was determined in HEK293T cells by MTT assay. The unloaded nanogels and the nanogel/pDNA complexes did not show significant cytotoxicity with over 85% cell viability compared to untreated (control) cells (Appendix A). To investigate the gene delivery performance of nanogels, first, the cellular uptake of nanogel/pDNA complexes was studied by CLSM and flow cytometry. Cy5-labeled pDNA (pDNA-Cy5) was complexed with NGs, NGs-MI, or NGs-BDD and incubated with HEK293T cells for 1 h and 4 h. As presented in Figure 4A, only a small amount of Cy5-fluorescence was associated with the cells at 1 h post-incubation. After prolonged incubation (4 h), a marked increase in cellular Cy5-fluorescence was observed for all different types of nanogel/pDNA complexes, with an uptake of NGs/pDNA > NGs-MI/pDNA > NGs-BDD/pDNA. This was confirmed by quantification of the cellular uptake of nanogel/pDNA-Cy5 complexes by means of flow cytometry (Figure 4B,C), indicating a time-dependent, efficient cellular uptake of the complexes by HEK293T cells. Partial colocalization of the pDNA-Cy5 signal with lysotracker, as observed by CLSM (Appendix A), indicated the presence of pDNA within late endosomes/lysosomes, suggesting that the nanogel/pDNA-Cy5 complexes entered the cells via endocytosis. The presence of gene delivery vectors in lysosomes is a well-known barrier for gene delivery, because of the presence of hydrolases that can degrade the delivery vectors and their genetic cargo. Therefore, escape of the genetic cargo from endo/lysosomes is a crucial factor for gene complexes in achieving efficient transfection. To investigate the endosomal escape mediated by nanogel/pDNA complexes, fluorescently labeled oligonucleotides (TAMRA-ODNs), i.e., a short DNA sequence, were used to visualize the intracellular release of genetic cargo. Because of the passive and rapid accumulation of ODNs in the cell nuclei following their release in the cell cytoplasm, fluorescent ODNs offer an easy way to investigate the endosomal escape of genetic cargo through quantification of the nuclear accumulation of ODNs following incubation of cells with nanogel/TAMRA-ODNs complexes. As shown in Appendix A, after 4 h incubation of HEK293T cells with fluorescent NGs/ODNs or NGs-MI/ODNs complexes, a punctate fluorescence pattern without fluorescent nuclei was observed, indicating the presence of the complexes within endosomes. In contrast, NGs-BDD/ODNs complexes and PEI-25kDa/ODNs complexes showed efficient endosomal escape with 48.5 ± 6.7 and 45.3 ± 11.2% of ODN-positive nuclei, respectively (Figure 5A). Branched PEI (25kDa), a well-known cationic polymer with high pH buffering capacity, is known to induce endosomal escape via the so-called proton sponge effect. However, the NGs-BDD nanogel showed no pH buffering capacity (Appendix A), implying the involvement of a pH-independent mechanism of cargo release from endosomes. To verify if the endosomal escape mediated by NGs-BDD was pH-independent, cells were incubated with NGs-BDD/ODNs in the absence and presence of bafilomycin A1, which inhibits the acidification of endosomes. Figure 5B shows that HEK293T cells incubated with NGs-BDD/ODNs in the presence of bafilomycin A1 revealed a similar number of ODN-positive nuclei as cells in the absence of bafilomycin A1. In sharp contrast, bafilomycin A1 completely inhibited the endosomal escape by PEI/ODNs complexes, as indicated by the absence of ODN-positive nuclei. Taken together, the data confirm that NGs-BDD/ODNs mediate the endosomal escape of ODNs via a pH-independent mechanism.

### 3.4. In Vitro Transfection with NGs-BDD/pDNA Complexes

Encouraged by their efficient cellular internalization and endosomal escape, we next investigated the transfection behavior of NGs-BDD/pDNA complexes. To this end, HEK293T cells were incubated with nanogel/pDNA complexes (pDNA encoding EGFP), and EGFP expression was qualitatively investigated by fluorescence microscopy. As shown in Figure 6A, EGFP expression in HEK293T cells, following incubation with NGs-BDD/pDNA, was comparable to that obtained with PEI-25kDa/pDNA complexes but lower than with Lipofectamine/pDNA. Cells incubated with NGs/pDNA and NGs-MI/pDNA complexes were essentially without EGFP expression. Quantification of EGFP expression by flow cytometry (Figure 6B) showed 58.9 ± 6.4 and 47.0 ± 5.1% transfection efficiency with NGs-BDD/pDNA and PEI-25kDa/pDNA, respectively. Transfection efficiency with Lipofectamine/pDNA was 89.0 ± 3.3%. Furthermore, in the presence of bafilomycin A1 (an inhibitor of endosomal acidifcation), PEI-mediated transfection showed a remarkable decrease of 81.7%, whereas transfection with NGs-BDD was only inhibited with 25.1% (Figure 6C), which correlates with the pH-independent endosomal escape of genetic cargo by the NGs-BDD nanogels (Figure 5B).

### 3.5. Membrane Perturbing Activity Is Responsible for NGs-BDD Mediated Transfection Capacity

The pH-independent transfection of NGs-BDD/pDNA complexes inspired us to investigate the membrane perturbing activity of the nanogels to see if their mechanism of action is similar to that of cationic lipids, i.e., if they directly destabilize the endosomal membrane through a membrane perturbing activity. For this purpose, we performed a hemolysis assay. In this assay, the release of hemoglobin from erythrocytes in the presence of a compound is taken as a measure of the membrane-perturbing activity of this compound. In Figure 7A, it can be seen that NG and NG-MI nanogels induced less than 10% hemolysis after 1 h of incubation, whereas NGs-BDD induced complete hemolysis. Furthermore, the percentage of hemolysis induced by NG-BDD/pDNA complexes was 78.2 ± 4.7%, while that of NG/pDNA and NG-MI/pDNA was below 20%. These data show that NGs-BDD exhibit a significant membrane-perturbing activity, which can explain for their high transfection capacity compared to NGs and NGs-MI.

## 4. Conclusions

In this study, a tertiary amine-functionalized nanogel (NGs) and quaternized nanogels with long aliphatic chains (NGs-BDD) and without long aliphatic chains (NGs-MI) were investigated for their gene delivery potential. We have demonstrated that NGs, harboring reversible charge, as well as positive charge-containing NGs-BDD and NGs-MI, can form complexes with pDNA and protect DNA from serum degradation. Moreover, the three nanogels exhibited similar cellular uptake and colocalization with lysosomes. However, only NGs-BDD demonstrated efficient endosomal escape and EGFP transfection in HEK293T cells, which occurred in a pH-independent way. We conclude that NGs-BDD mediate endosomal escape of genetic cargo through a membrane-perturbing activity, which can be attributed to the presence of long aliphatic chains in NGs-BDD. This cationic lipid-like mechanism of transfection by the NGs-BDD nanogel offers opportunities for the rational design and improvement of nanogel-based gene delivery vectors to improve non-viral gene therapy of diseases. Future research should focus on nanogel biocompatibility, time-controlled cargo release, and in vivo biodistribution, which is vital for the development of non-viral gene vectors toward future clinical applications.

## Figures and Tables

**Figure 1 pharmaceutics-13-01964-f001:**
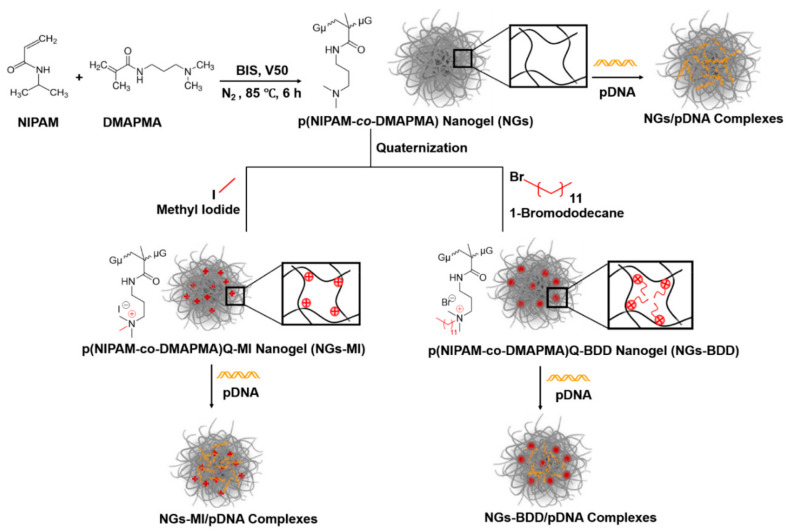
Reaction scheme for NGs, NGs-MI, NGs-BDD, and their complexes with DNA.

**Figure 2 pharmaceutics-13-01964-f002:**
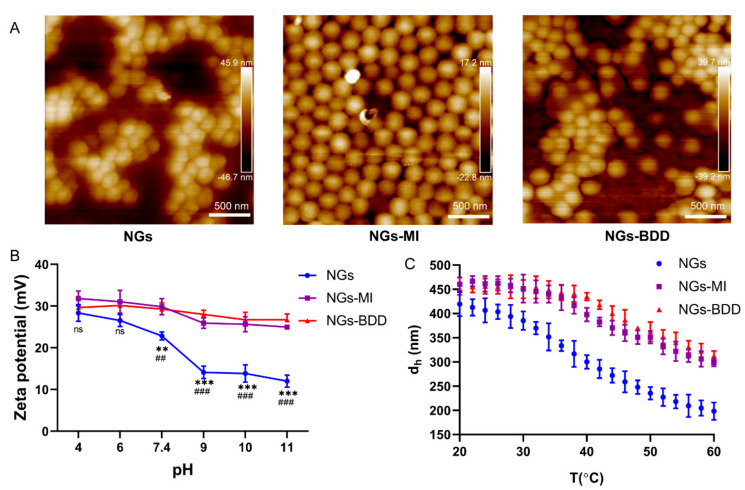
Characterization of NGs, NGs-MI, and NGs-BDD nanogels. (**A**) Atomic force microscopy (AFM) images of nanogels in the dry state at room temperature. (**B**) pH-dependent zeta potential of nanogels in 0.05 M NaCl at 25 °C. (*n* = 3; ** *p* and ^##^ *p* < 0.005, *** *p* and ^###^ *p* < 0.001 with * indicating comparison with NGs, ^#^ indicating comparison with NGs-MI). (**C**) Temperature-dependent hydrodynamic diameter of nanogels in water.

**Figure 3 pharmaceutics-13-01964-f003:**
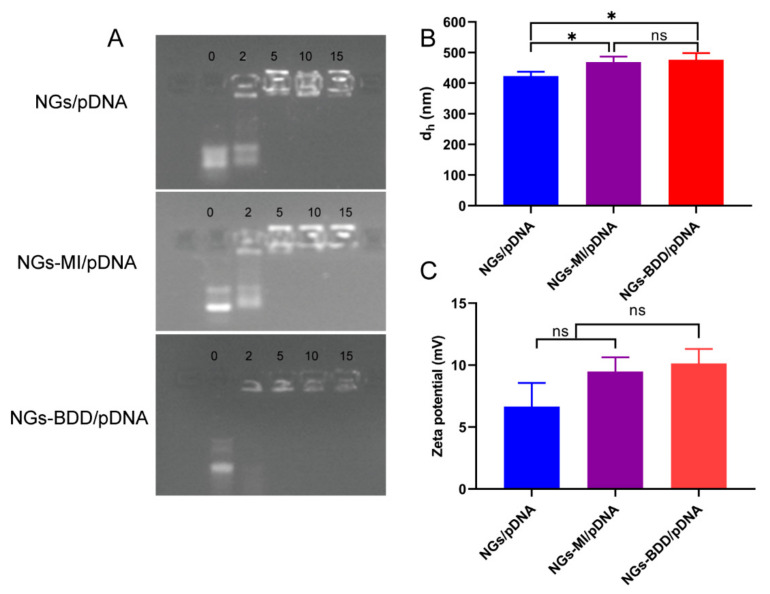
(**A**) Agarose gel electrophoresis of the complexes of nanogels and pDNA prepared at weight ratios (*w*/*w*) from 0–15, at room temperature. (**B**) Hydrodynamic diameter and (**C**) Zeta potential analysis of nanogel/pDNA complexes prepared at a weight ratio of 10, at room temperature (*n* = 3, * *p* < 0.05).

**Figure 4 pharmaceutics-13-01964-f004:**
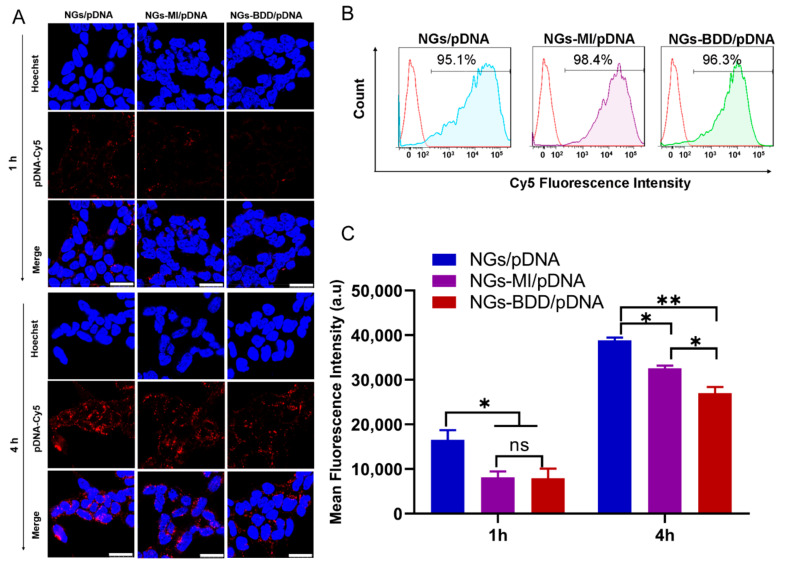
Cellular uptake of nanogel/pDNA complexes in HEK293T cells. (**A**) CLSM images of cells after incubation with NGs/pDNA, NGs-MI/pDNA, and NGs-BDD/pDNA complexes for 1 h and 4 h. pDNA and nuclei were labeled with Cy5 and Hoechst 33342, respectively. Scale bars are 25 μm. (**B**) Flow cytometry histograms of cells incubated with nanogel/pDNA-Cy5 complexes for 4 h. (**C**) Mean fluorescence intensities of cells incubated with nanogel/pDNA-Cy5 complexes at indicated time points. Data presented mean ± SD from three independent experiments performed in duplicate (* *p* < 0.05, ** *p* < 0.005).

**Figure 5 pharmaceutics-13-01964-f005:**
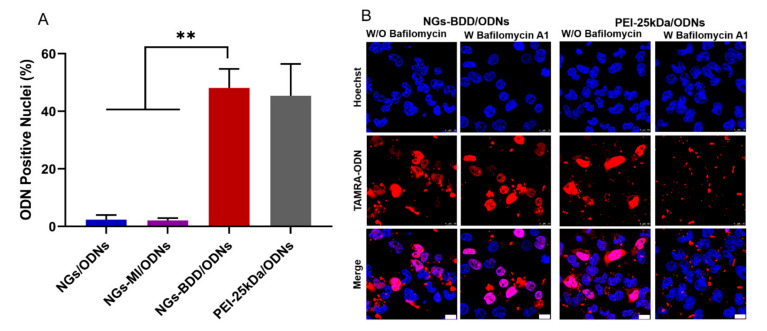
Endosomal escape of TAMRA-labeled ODNs (red) mediated by nanogels and PEI-25kDa in HEK293T cells. (**A**) Semi-quantitative analysis (% cells with ODN positive nuclei) of CLSM images. Data present mean ± SD of three independent experiments and 150–200 cells (9 images in total) per condition (** *p* < 0.005). (**B**) CLSM images of cells after incubation with NGs-BDD/ODNs and PEI-25kDa/ODNs for 4 h with or without Bafilomycin A1. The cell nuclei were stained with Hoechst33324 (blue). Scale bars are 10 μm.

**Figure 6 pharmaceutics-13-01964-f006:**
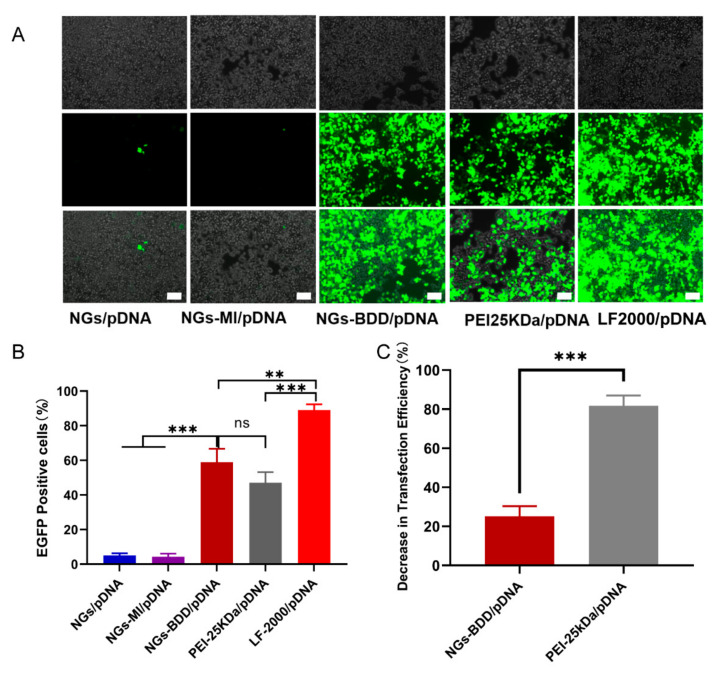
Transfection efficiency with nanogel/pDNA PEI/pDNA and LF2000/pDNA complexes in HEK293T cells. (**A**) Light microscopy images of nanogel-treated HEK293T cells (upper row: phase contrast; middle row: fluorescence; lower row: merged phase contrast and fluorescence. Scale bars indicate 100 μm). (**B**) Percentage of EGFP-positive cells measured by flow cytometry after treatment with complexes for 48 h. (**C**) Decrease in percentage of EGFP-positive cells after treatment with NG-BDD/pDNA complexes and PEI-25kDa/pDNA complexes in the presence of Bafilomycin A1 compared to treatment in the absence of Bafilomycin A1. Data present mean ± SD of three independent experiments performed in duplicate (** *p* < 0.005, *** *p* < 0.001).

**Figure 7 pharmaceutics-13-01964-f007:**
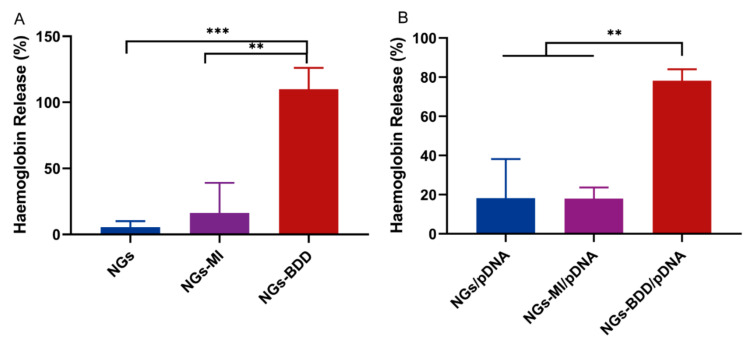
Membrane-perturbing activity of nanogels and their complexes with pDNA. Erythrocytes were incubated with (**A**) NG, NG-MI, and NG-BDD nanogels and (**B**) complexes, and hemoglobin release was measured by absorption spectroscopy. 100% hemoglobin release was induced with 1% Triton X-100 (*n* = 3, ** *p* < 0.005 *** *p* < 0.001).

## Data Availability

All data are included in this manuscript and its Appendix A.

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
