# Peer review of "Aliphatic Quaternary Ammonium Functionalized Nanogels for Gene Delivery"

_pharmaceutics, 2021, doi:10.3390/pharmaceutics13111964_

Round 1
Reviewer 1 Report
The paper liphatic quaternary ammonium functionalized nanogels for gene delivery, prepared by Huaiying Zhang et al., presents novel and interesting results that deserve to be published after some minor improvements:
- font size of figure 1 must be increased.
- Same issue in figure 4-B
- Figure 5 - there are figures where the scale bar is missing.
- Figure 4 - same issue.
- Figure 6 - same issue
Reviewer 2 Report
The manuscript entitled “Aliphatic quaternary ammonium functionalized nanogels for gene delivery”, by Zhang et al have prepared and characterized poly(N-iso-propylacrylamide) nanogels with varying cationic functionalities and tested them in vitro in HEK293T cell culture for cytotoxicity, cellular uptake, transfection, and mechanism of cellular escape. The research is well designed and executed. The data supports the conclusions obtained. Although the work is very preliminary, it is intriguing. There are a few minor comments for the authors as noted below.
Comments to authors:
- Authors should perform additional experiments with well acknowledged delivery systems like lipid nanoparticles, lipofectamine etc., for comparative understanding of their delivery system’s efficacy. If not, authors must at-least include these experiments in limitations and future directions/studies section as part of the conclusion.
- Include statistical analysis for data presented in figures 2, 3 and 4.
